# Genomics and Antimicrobial Susceptibility of Clinical *Pseudomonas aeruginosa* Isolates from Hospitals in Brazil

**DOI:** 10.3390/pathogens12070918

**Published:** 2023-07-08

**Authors:** Carlos Henrique Camargo, Amanda Yaeko Yamada, Andreia Rodrigues de Souza, Marisa de Jesus de Castro Lima, Marcos Paulo Vieira Cunha, Pedro Smith Pereira Ferraro, Claudio Tavares Sacchi, Marlon Benedito Nascimento dos Santos, Karoline Rodrigues Campos, Monique Ribeiro Tiba-Casas, Maristela Pinheiro Freire, Pasqual Barretti

**Affiliations:** 1Centro de Bacteriologia, Instituto Adolfo Lutz, Sao Paulo 01246-902, SP, Brazil; amandayy45@gmail.com (A.Y.Y.); andreiarsouza1970@gmail.com (A.R.d.S.); marisalima119@gmail.com (M.d.J.d.C.L.); cunha.mpv@gmail.com (M.P.V.C.); pedrosp.ferraro@outlook.com (P.S.P.F.); monique.casas@ial.sp.gov.br (M.R.T.-C.); 2Faculdade de Medicina, Universidade de São Paulo, Sao Paulo 01246-902, SP, Brazil; maristelapf@uol.com.br; 3Laboratório Estratégico, Instituto Adolfo Lutz, Sao Paulo 01246-902, SP, Brazil; labestrategico@ial.sp.gov.br (C.T.S.); marlon.beneditto@gmail.com (M.B.N.d.S.); karol_rodriguescamp@yahoo.com.br (K.R.C.); 4Faculdade de Medicina de Botucatu, Universidade Estadual Paulista, Botucatu 18618-686, SP, Brazil; pasqual.barretti@unesp.br

**Keywords:** CAZ-AVI, MEM-VAR, cefiderocol, fosfomycin, polymyxin, whole genome sequencing, Illumina, MLST

## Abstract

*Pseudomonas aeruginosa*, an opportunistic pathogen causing infections in immunocompromised patients, usually shows pronounced antimicrobial resistance. In recent years, the frequency of carbapenemases in *P. aeruginosa* has decreased, which allows use of new beta-lactams/combinations in antimicrobial therapy. Therefore, the in vitro evaluation of these drugs in contemporary isolates is warranted. We evaluated the antimicrobial susceptibility and genomic aspects of 119 clinical *P. aeruginosa* isolates from 24 different hospitals in Brazil in 2021–2022. Identification was performed via MALDI-TOF-MS, and antimicrobial susceptibility was identified through broth microdilution, gradient tests, or disk diffusion. Whole-genome sequencing was carried out using NextSeq equipment. The most active drug was cefiderocol (100%), followed by ceftazidime–avibactam (94.1%), ceftolozane–tazobactam (92.4%), and imipenem–relebactam (81.5%). Imipenem susceptibility was detected in 59 isolates (49.6%), and the most active aminoglycoside was tobramycin, to which 99 (83.2%) isolates were susceptible. Seventy-one different sequence types (STs) were detected, including twelve new STs described herein. The acquired resistance genes *bla*_CTX-M-2_ and *bla*_KPC-2_ were identified in ten (8.4%) and two (1.7%) isolates, respectively. Several virulence genes (*exoSTUY, toxA, aprA, lasA/B, plcH*) were also identified. We found that new antimicrobials are effective against the diverse *P. aeruginosa* population that has been circulating in Brazilian hospitals in recent years.

## 1. Introduction

*Pseudomonas aeruginosa* ranks as the most frequent non-fermentative Gram-negative bacterium associated with hospital-acquired infections, mainly affecting seriously ill patients in intensive care units (ICUs) [1]. It is part of the ESKAPE group, which comprises drug-resistant pathogens of clinical importance, namely *Enterococcus*, *Staphylococcus aureus*, *Klebsiella pneumoniae*, *Acinetobacter baumannii*, *P. aeruginosa*, and *Enterobacter* [2]. *P. aeruginosa* presenting carbapenem resistance is listed as one of the critical pathogens defined by the World Health Organization and as a serious threat according to the Centers for Disease Control and Prevention [3,4]. In the United States, the rate of hospital-onset multidrug-resistant (MDR) *P. aeruginosa* infections increased by more than 30% in 2020 compared to 2019, mainly due to the COVID-19 pandemic [5]. In Brazil, high endemic rates of drug-resistant pathogens are reported [6,7]. Regarding *P. aeruginosa* specifically, the SPM-1-producing, colistin-only susceptible (COS) ST277 clone has been reported to predominate in the country throughout the last two decades [8]. Over the years, surveillance studies have shown stability in the rates of carbapenem-resistance in *P. aeruginosa* from Latin America [9]. Nevertheless, the frequency of SPM-producing *P. aeruginosa* has been consistently reported to be decreasing in recent years [10,11,12], showing the involvement of non-enzymatic mechanisms in carbapenem resistance, including porin loss and efflux pump system overproduction [13]. This change in the epidemiology of carbapenem resistance allows for the opportunity to use new beta-lactams/combinations in the treatment of *P. aeruginosa* infections. Therefore, it is necessary to perform an in vitro evaluation of these drugs using contemporary isolates from clinical sources. In this study, we aimed to evaluate a large collection of clinical *P. aeruginosa* isolates recovered from various hospitals in Brazil during 2021–2022. The whole-genome sequence of the isolates was obtained in order to define their clonality, resistome, and virulome characteristics.

## 2. Materials and Methods

### 2.1. Isolate Selection and Identification

On a continuous and voluntary basis, the Instituto Adolfo Lutz receives clinical isolates of hospitalized patients presenting infections, associated with outbreaks or not, for phenotypic and genotypic antimicrobial resistance characterization. Bacterial identification was initially carried out via phenotypic testing and MALDI-TOF MS (Bruker Daltonics, Bremen, Germany). Between January 2021 and August 2022, 216 isolates identified as *P. aeruginosa* were received in our laboratory. A total of 70 isolates recovered from non-human sources (hospital environment) or redundant isolates from the same patient (recovered within one month) were excluded; thus, 146 isolates were analyzed.

### 2.2. Antimicrobial Susceptibility Testing

The isolates were initially processed using disk-diffusion antimicrobial susceptibility testing for amikacin, aztreonam, cefepime, ceftazidime, ciprofloxacin, doripenem, gentamicin, imipenem, levofloxacin, meropenem, netilmicin, piperacillin-tazobactam, ticarcillin-clavulanate, and tobramycin with Oxoid (Basingstoke, United Kingdom) disks. Complementarily, an in-house broth microdilution method using cation-adjusted Mueller–Hinton Broth (Sigma-Aldrich, St. Louis, MO, USA) was performed to determine the minimum inhibitory concentration (MIC) for amikacin gentamicin, imipenem, meropenem, colistin, polymyxin B, tigecycline, and ceftazidime-avibactam, and all salts were purchased from Sigma-Aldrich (St. Louis, MO, USA), except for avibactam, which was donated by Pfizer Inc. To complete the antimicrobial susceptibility panel, novel antimicrobials/combinations were evaluated with Liofilchem (Roseto degli Abruzzi, Italy) MIC test strips for ceftolozane–tazobactam, meropenem–vaborbactam, imipenem–relebactam, cefoperazone–sulbactam, cefiderocol, plazomicin, eravacycline, and fosfomycin. 

### 2.3. Phenotypic and Genotypic Carbapenemase Detection

We used a Fourier test to detect the production of carbapenemases [14]. This test consists of applying different amounts of cloxacillin salt in imipenem disks to discriminate between carbapenemase-producing and non-producing strains of *P. aeruginosa*. Isolates presenting differences in imipenem halos with and without cloxacillin ≤ 5 mm were identified as carbapenemase producers. We excluded isolates producing metallo-β-lactamases (MBL) detected via multiplex PCR targeting the main genes *bla*_NDM_, *bla*_SPM_, *bla*_IMP_, and *bla*_VIM_ [15].

### 2.4. DNA Extraction, Whole Genome Sequencing, and Assembly

Whole bacterial DNA was extracted by using Invitrogen PureLink Genomic DNA Mini Kit (USA) following the manufacturer’s recommendations. After extraction, the DNA was quantified using a Qubit (Thermo Scientific Inc., Waltham, MA, USA) fluorometer and the libraries were prepared for Illumina (USA) NextSeq sequencing, using a P1/300 cycle cartridge. The library preparation and Illumina runs were performed at the Strategic Laboratory, Instituto Adolfo Lutz, São Paulo, Brazil. After the evaluation of quality control parameters (read sizes, Phred values > 30, GC content), genomes were de novo assembled using CLC Workbench software (Qiagen, Germany). 

### 2.5. Annotation, Resistome and Virulome Detection, Serotype Prediction, and MLST

The assembled genomes were uploaded to the Galaxy Europe platform [16] and then annotated with Prokka [17]. Acquired resistance and virulence-codifying genes were detected with Abricate using the Resfinder [18] and VirulenceFinder [19] databases, respectively. Chromosomal mutations leading to antimicrobial resistance were detected using Pointfinder software [20]. The in silico serotype was determined via the pseudomonas aeruginosa serotyper (PAst) program [21] available at the Center for Genomic Epidemiology webserver (https://www.genomicepidemiology.org/, accessed on 22 April 2023). Sequence types (STs) were defined based on the internal sequences of seven housekeeping genes [22]. When new alleles or allele combinations were identified, the isolates were submitted to pubMLST for curation and assignment [23].

### 2.6. Phylogenetic Analysis

The Prokka-annotated genomes were used to generate a core genome alignment with Roary v3.13.0 [24]. Core genome phylogeny was inferred from the core genome alignment, and a maximum-likelihood tree was constructed using IQ-TREE [25] v.2.1.2 with standard model selection followed by tree inference and 1000 bootstrap replicates. The tree was visualized in the Microreact platform [26] and can be interactively accessed at https://microreact.org/project/ogubUpsSXzEseq311xMeth-pseudomonas-non-mbl-119 (accessed on 22 April 2023). 

## 3. Results

Initially, 146 isolates were enrolled in this study (75 of them were imipenem resistant), but 12 were excluded because they were MBL-producers (MBL frequency among the imipenem-resistant isolates was 16%), namely 6 *bla*SPM, 4 *bla*VIM, and 2 *bla*IMP isolates. In addition, quality parameters were not achieved after the sequencing of 15 isolates, which were excluded. Therefore, 119 isolates recovered from the clinical specimens of 115 non-repetitive patients who attended one of twenty-four public or private health institutions located in 16 different Brazilian cities, were included. The isolates were recovered mainly from the respiratory tract (n = 58; 48.7%), followed by blood or central venous catheter tips (26; 21.8%), urine (16; 13.4%), infected wounds (6; 5.0%), surveillance swabs (3; 2.5%), or cerebrospinal fluid (1, 0.8%), and nine isolates (7.6%) were recovered from other clinical sources. 

Antimicrobial susceptibility evaluated via disk-diffusion is presented in Table 1. According to the Magiorakus classification, most of the isolates were identified as multidrug-resistant (MDR) (55.5%) or extensively drug-resistant (XDR) (18.5%), and one isolate (0.8%) was identified as pan-drug-resistant (PDR). The remaining 30 isolates (25.2%) were classified as susceptible. 

MIC values determined via broth-microdilution or gradient tests (Table 2) showed that all isolates were susceptible to cefiderocol (100% susceptibility), and the most active drugs were ceftazidime–avibactam (94.1%), ceftolozane–tazobactam (92.4%), and imipenem–relebactam (81.5%). Among classical drugs, a comparable susceptibility rate was found only for tobramycin (83.2%). The carbapenem susceptibility rates ranged from 47.9% (imipenem, broth microdilution) to 66.4% (doripenem, disk-diffusion). Resistance to colistin and polymyxin B was identified in four (3.4%) and three (2.5%) isolates, respectively; according to CLSI, the remaining isolates were classified as intermediate (I). The distribution of MIC values, as well as the MIC50/MIC90 values, demonstrated the inefficacy of tigecycline, fosfomycin, and eravacycline against these *P. aeruginosa* isolates (Table 2). 

Genomic analysis identified 71 different sequence types (STs) among the 119 *P. aeruginosa* sequences evaluated. Of note, twelve new STs were identified for the first time in this study, of which eight are new allele combinations and three are new alleles (*mutL*, *aroE*, *trpE*)—one isolate presented new alleles for two genes simultaneously (*guaA* and *mutL*). The most frequent ST, ST235, was found in sixteen isolates (13.4%) from eight hospitals across seven cities; this ST was identified in isolates recovered from different sources, including blood, infected wounds, respiratory tracts, and surveillance swabs. The distribution of common STs (detected in more than one isolate), according to hospital and isolation source, is presented in Table 3. An in silico analysis identified eleven predicted serotypes, with O11 (33.6%) and O1 (10.1%) as the most frequent, and all the ST235 isolates (n = 16) presented the O11 predicted serotype (Appendix A). The phylogenetic tree based on pangenome analysis correlated with STs and partially with the predicted serotypes, showing that the ST235 isolates to be more resistant than isolates with other STs (Figure 1; Appendix A). In fact, the XDR phenotype was more frequent in ST235 isolates (12/16, 75%) than in isolates belonging to other STs (10/103, 10.7%) (*p* < 0.0001 on the Yates’ chi-squared test).

In addition to the chromosomally and naturally encoded β-lactamases *ampC* and *bla*_OXA_, acquired antimicrobial resistance genes conferring resistance to beta-lactams were identified. Nine isolates (eight with the XDR phenotype and one MDR) carried the ESBL *bla*_CTX-M-2_ gene, and they were identified as ST235 (n = 6), ST111, ST244, and ST309 (n = 1, each). The *bla*_GES-1_ gene was identified in another ST235 XDR isolate, and the *bla*_KPC-2_ gene was found in one XDR isolate (ID_0367_21) belonging to ST803. One isolate (ID_0455_22), XDR, ST1284, simultaneously carried the *tetG*, *bla*_KPC-2_ and *bla*_CTX-M-2_ genes. The analysis of the genetic environment associated with the *bla*_KPC-2_ gene showed that isolate ID_0367_21 carries the gene on a plasmid with a DNA sequence similar to plasmids harboring carbapenemases that circulate in *P. aeruginosa* worldwide (corresponding to GenBank Accession Numbers CP078000, LC586269, CP077989). On this plasmid, the gene is located in a truncated non-*Tn4401* genetic element (NTEKPC), similar to that presented in FII-FIB(pQil) plasmids that harbor KPC-3 in *Klebsiella pneumoniae* isolates from the USA [27]. In contrast, it was not possible to define whether the *bla*_KPC-2_ gene of isolate ID_0455_22 is present on a plasmid or the chromosome. However, *bla*_KPC-2_ is present in a truncated form of Tn*4401* element isoform *b*. Our analysis revealed that a segment of 6844 bp with 99.9% similarity to Tn*4401b* and an intact IRR is present in the genome of this isolate. Another MDR isolated with ST309 was found to carry the *tetG* gene. Several genes that confer resistance to aminoglycosides were identified, such as *aac*, *aad*, *ant*, and *aph*; conversely, *armA* or *rmt*-family genes, which are associated with high levels of aminoglycoside resistance, were not detected.

Genes associated with the four effector proteins of the type III secretion system (T3SS), namely *exoS, exoT, exoU*, and *exoY*, were detected at frequencies of 63.8%, 98.3%, 30.2%, and 92.4%, respectively. As expected [28], the association of *exoS* and *exoU* in the same isolate was not detected. Additionally, genes encoding exotoxin A (*toxA*) were detected in 97.5% of the isolates; alkaline protease (*aprA*), elastase (*lasA/B*), and phospholipase C (*plcH*) were detected in all of the isolates (except for one isolate negative for the *lasA* gene).

## 4. Discussion

Although studies have focused on changes in *P. aeruginosa* antimicrobial susceptibility, only scarce information is available for genomics findings for this species, particularly for specimens with the MBL-negative phenotype, which is increasing in Brazilian settings.

In this study, we identified that antimicrobial agents not in use in clinical settings have preserved activity against clonally diverse *P. aeruginosa* clinical isolates. *P. aeruginosa* is recognized as an opportunistic pathogen causing skin and soft tissue infections and even potentially fulminant invasive infections [29]. The treatment of *P. aeruginosa* infections is limited because of both intrinsic and acquired resistance mechanisms [30]. Carbapenems have been employed for clinical treatment, but with the emergence of metallo-beta-lactamases, a potent class of carbapenemases, beta-lactams (with the exception of aztreonam, which is not hydrolyzed by MBL) are no longer recommended for the treatment of *P. aeruginosa* [30]. It is known that difficult-to-treat infections due to *P. aeruginosa* result in worse prognosis with poor outcomes [1].

Molecular studies have identified that the spread of MBL production in Brazil is associated with a dissemination of a specific sequence type, ST277, the so-called colistin-only susceptible (COS) *P. aeruginosa* [31]. More recently, however, a reduction in the prevalence of MBL-producing ST277 was observed in Brazilian hospitals, but the reasons for this change in epidemiology are not clear [10,11,12]. By analyzing a large collection of *P. aeruginosa* recovered from the clinical specimens of patients admitted to different hospitals, we identified 71 different STs among the 119 isolates evaluated. The diversity of clones, which has been reported in previous smaller studies in recent years, may be linked to the widespread misuse of antimicrobials rather than clonal dissemination through cross-transmission [32].

Intriguingly, we identified that ST235 isolates carried more resistance markers than isolates with other STs. ST235 is considered a “high-risk clone” because of its ability to accumulate antimicrobial resistance genes and is widespread in diverse hospital settings [33]. We speculate that a transition from the ST277 to the ST235 *P. aeruginosa* clone, which is still expanding in Brazilian hospitals, has been occurring. As ST235 was found to display more resistance than the other clones (Figure 1), attention must be paid to this high-risk clone to avoid or at least reduce its dissemination.

We identified the high activity of cefiderocol (100%), ceftazidime–avibactam (94.1%), ceftolozane–tazobactam (92.4%), and imipenem–relebactam (81.5%) in our analyses. Of these drugs, only cefiderocol is currently not approved for the treatment of *P. aeruginosa* infections in Brazil. Avibactam and relebactam are second-generation beta-lactamase inhibitors able to inhibit class A, C, and D beta-lactamases, with activity against KPC-producing Enterobacterales and KPC-producing *Pseudomonas* [34]. In our study, two isolates with KPC-2 in different genetic backgrounds were identified, which is not common in our region [35] but has been reported in other countries [36,37,38,39].

Regarding currently available drugs in hospital settings, polymyxin B and colistin were found to present high activity against the *P. aeruginosa* isolates, as observed by the MIC50 and MIC90 results (Table 2). In a realistic scenario, empiric treatment with polymyxins is still employed in Brazilian settings, which, in theory, would cover the contemporary *P. aeruginosa* isolates causing hospital-acquired infections in the country. Nevertheless, recent guidelines do not recommend use of this class of drugs for severe infections caused by carbapenem-resistant *P. aeruginosa*, as some studies have shown worse outcomes, especially when compared to ceftolozane–tazobactam [40]. Molecular analysis identified that only 16% of the carbapenem resistance in *P. aeruginosa* is mediated by metallo-beta-lactamase production, rendering the use of ceftolozane–tazobactam, ceftazidime–avibactam, or other beta-lactam combinations instead of polymyxins possible as empirical treatments in Brazilian hospitals.

In addition to the robust data on antimicrobial susceptibility generated in this study, the determination of the resistome and virulome for each isolate is notable. The inclusion of such genomes in public databases will contribute to future studies with both local and global perspectives, allowing for the tracking of an important public health pathogen of clinical relevance.

## 5. Conclusions

In summary, by analyzing 119 *P. aeruginosa* isolates with diverse genetic backgrounds from 24 Brazilian hospitals, we identified the preserved activity of antimicrobial agents with restricted its use in our country. The continuous investigation of antimicrobial susceptibility, associated with the rational use of antimicrobial agents in clinical practice, is essential to preserving the scarce options for the treatment of bacterial pathogens causing health care-associated infections.

## Figures and Tables

**Figure 1 pathogens-12-00918-f001:**
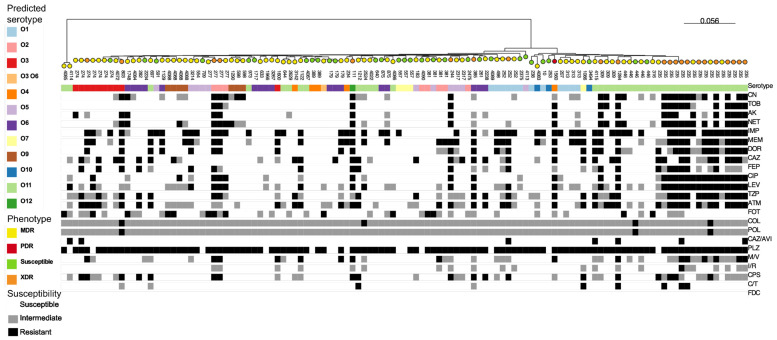
Maximum-likelihood tree generated through core genome alignment with Roary followed by IQ-TREE analysis in 119 isolates (Appendix A) of clinical *P. aeruginosa* from Brazil. In this tree, the ST (indicated by the number for each strain), predicted serotype (colored squares), antimicrobial resistance (white, gray, or black squares), and resistance phenotypes (colored circles) are represented. The tree can be interactively accessed on the link https://microreact.org/project/ogubUpsSXzEseq311xMeth-pseudomonas-non-mbl-119 (accessed on 22 April 2023).

**Table 1 pathogens-12-00918-t001:** Antimicrobial susceptibility of clinical isolates of *P. aeruginosa* from Brazil recovered between 2021 and 2022 (n = 119).

Antimicrobial Category	Antimicrobial Agent	Susceptible		Intermediate		Resistant	
		n	%	n	%	n	%
Aminoglycosides	Amikacin	93	78.2	7	5.9	19	16.0
	Gentamicin	78	65.5	17	14.3	24	20.2
	Tobramycin	99	83.2	2	1.7	18	15.1
	Netilmicin	83	69.7	10	8.4	26	21.8
Antipseudomonal carbapenems	Imipenem	59	49.6	3	2.5	57	47.9
	Meropenem	67	56.3	9	7.6	43	36.1
	Doripenem	79	66.4	10	8.4	30	25.2
Antipseudomonal cephalosporins	Cefepime	86	72.3	14	11.8	19	16
	Ceftazidime	75	63.0	12	10.1	32	26.9
Antipseudomonal fluoroquinolones	Ciprofloxacin	84	70.6	9	7.6	26	21.8
	Levofloxacin	63	52.9	26	21.8	30	25.2
Antipseudomonal penicillins + β-lactamase inhibitors	Ticarcillin–clavulanic acid	0	0	20	54.1	17	45.9
	Piperacillin–tazobactam	65	54.6	24	20.2	30	25.2
Monobactam	Aztreonam	54	45.4	28	23.5	37	31.1
Phosphonic acids	Fosfomycin	77	64.7	28	23.5	14	11.8
Polymyxins	Polymyxin B ^1^	0	0	116	97.5	3	2.5
	Colistin ^1^	0	0	115	96.6	4	3.4

^1^ determined via broth microdilution.

**Table 2 pathogens-12-00918-t002:** Antimicrobial susceptibility (quantitative results) of *Pseudomonas aeruginosa* isolates from Brazil (n = 119). The table indicates the breakpoints for susceptibility categorization and the reference from which those breakpoints were retrieved. The minimal inhibitory concentration (MIC) values (μg/mL) for inhibiting the growth of 50% (MIC50, **bold**) and 90% (MIC90, underlined) of the isolates are indicated, as is the cumulative distribution of isolates.

Drug ^Method^	Breakpoint (S | R)	Reference	% of			Number of Isolates Inhibited at Minimal Inhibitory Concentration (MIC, in μg/mL) Indicated Below											
			S	I	R	0.06	0.125	0.25	0.5	1	2	4	8	16	32	64	>64
Amikacin ^1^	≤16 | ≥64	CLSI 2022	75.6	9.2	15.1				1	3	4	33	**69**	90	101	105	119
Gentamicin ^1^	≤4 | ≥16	CLSI 2022	59.7	16.0	24.4		1		3	6	29	**71**	90	100	107		119
Imipenem ^1^	≤2 | ≥8	CLSI 2022	47.9	7.6	44.5			3	20	48	57	**66**	84	104	111	117	119
Meropenem ^1^	≤2 | ≥8	CLSI 2022	50.4	10.1	39.5	1	11	31	42	49	**60**	72	85	108	116	118	119
Colistin ^1^	≤2 = I | ≥4	CLSI 2022	NA	96.6	3.4	3	5	12	57	**106**	115	119					
Polymyxin B ^1^	≤2 = I | ≥4	CLSI 2022	NA	97.5	2.5		3	16	**73**	116		118	119				
Tigecycline ^1^	NA	NA	NA	NA	NA					1	2	3	29	**86**	113	118	119
Ceftazidime-avibactam ^1^	≤8/4 | ≥16/4	CLSI 2022	94.1	0	5.9				2	30	**73**	100	112	116	118	119	
Plazomicin ^2^	S ≤ 2	U.S. FDA for Enterobacterales	11.8	0	88.2		1	2	3		14	55	**83**	108	112	114	119
Meropenem-vaborbactam ^2^	≤4/8 | ≥16/8	CLSI 2022 for Enterobacterales	66.4	20.2	13.4	7	19	36	44	51	**61**	79	95	114	116	117	119
Imipenem–relebactam ^2^	≤2/4 | ≥8/4	CLSI 2022	81.5	17.6	0.8			1	27	**63**	97	117	119				
Ceftazidime–avibactam ^2^	≤8/4 | ≥16/4	CLSI 2022	89.9	4.2	5.9			1	3	25	**61**	94	107	113	116	119	
Cefoperazone–sulbactam ^2^	≤16 | ≥64	Sulperazone^®^ package insert	64.7	19.3	16.0						19	47	**66**	77	90	104	119
Ceftolozane–tazobactam ^2^	≤4/4 | ≥16/4	CLSI 2022	92.4	2.5	5.0			1	22	**73**	96	110	112	113	114		119
Cefiderocol ^2^	≤4 | ≥16	CLSI 2022	100.0	0	0	**66**	92	105	115	117	118	119					
Fosfomycin ^2^	NA	NA	NA	NA	NA						2	6	7	10	15	29	**119**
Eravacycline ^2^	NA	NA	NA	NA	NA				1		2	14	23	34	46		**119**

^1^ BMD; in house broth-microdilution. ^2^ Gradient strips; if necessary, MIC values were rounded up to the next log2 dilution. NA = not applicable.

**Table 3 pathogens-12-00918-t003:** Distribution of common ST (>1 isolate) clinical isolates of *P. aeruginosa* from Brazil recovered between 2021 and 2022 (n = 119), according to municipalities, hospitals, and source of isolation.

ST	Number of Isolates	Number of Cities	Number of Hospitals	Source *
235	16	7	8	B I O R S
274	7	3	4	C R U
446	5	2	2	B I O R
277	4	2	4	O R U
381	4	2	2	B R
179	3	2	2	R
252	3	1	1	B R
309	3	2	2	B R U
313	3	3	3	R U
557	3	1	1	B R U
389	2	1	1	B
598	2	1	1	B U
875	2	1	1	I R
2317	2	1	1	B R

* B = blood or central venous catheter tip; C = cerebrospinal fluid; I = infected wound; O = other clinical sources; R = respiratory tract; S = surveillance swabs; U = urine.

## Data Availability

All sequences generated in this study were deposited in pubMLST and NCBI/Genbank platforms (Appendix A).

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
