# Peer review of "Genomics and Antimicrobial Susceptibility of Clinical Pseudomonas aeruginosa Isolates from Hospitals in Brazil"

_pathogens, 2023, doi:10.3390/pathogens12070918_

Round 1
Reviewer 1 Report
I find this article interesting, concisely written and clearly highlight key findings.
However, few things needed to be addressed to make it perfect.
Even though I don't feel qualified to correct the English, I detect some direct translations and I think some sentences needed to be rephrased. For example,
Line 13-14 “In recent years, the incidence of carbapenemases in P. aeruginosa has decreased, opening the opportunity for the use of new betalactams/combinations”
How is the decrease of carbapenemases in P. aeruginosa has open opportunity for using new betalactams? Do you mean “increase” ?
Line 41-42: please specify the rates and years
Can the authors provide details about the “decrease” in Brazil? World? (Incidence rate and years)
Line 59: “hospital acquired infections” is it clonally demonstrated by genomics of free statement?
Line 65: “recovered up to one month apart” do you mean isolated withing a month? Please rephrase it.
I usually recommend a sample selection diagram: total number of samples, how many selected and or excluded based on criteria etc.. it maked it easier to follow.
Line 68: “submitted”, I suggest “process using”
Line 82: “employed”, I suggest “used”
I recommend avoiding passive voice, active voice is usually recommended for scientific article or report: we, I etc…
Line 114: interesting interactive plot, but I can’t see node label legend.
Line 120: “isolates did not achieve quality parameters after sequencing and were excluded” you mean “quality performance” instead of “quality parameters”? I agree with the quality issue but I think you can go around it by doing core genome alignment directly for your fastqs using snippy ​​https://github.com/tseemann/snippy , please check.
Line 131: how do you define MDR and XDR for P.aeruginosa? When the author say “or” with two different distributions, it is not clear to me.
Line 159: intrinsic blaOXA genes: do you mean chromosomic? Have you checked if there were any recombination marquers around? IS? Transposons? Ligase? etc..
How do you distinguish between acquired and chromosomic genes? Please provide comparative view with easyfig or artemis ACT or any other.
Please provide tip node color legend, I think it is important to pinpoint the clonal distribution and match it with the ST.
Reviewer 2 Report
Line 84: ... AmpC hyperproducing ,BUT porin loss ... please, check statement.
Lines 86-88: Ref. 15 describes the relevance of carbapenemases in 2011, including the classification of “main (MBL) carbapenemases” in P.aeruginosa such as IMP, VIM, SPM, which are excluded from analysis. Why are other MBLs not included which may be more relevantt in 2020?
Tables 1 and 2: What was the reason to use two different methods for susceptibility testing? Please, provide an explanation.
Line 159: What is meant: “intrinsic blaOXA genes..”? (compare to ampC as chromosomally encoded beta-lactamase gene.
Tables 1 and 2 as well as figure 1 are partially redundant. Please, check, if it is possible to combine information.
Line 216: What is meant: “in vitro phenotype..”?
Line 235: What is meant: “more resistant..”
Round 2
Reviewer 2 Report
There are still a few points to be modified/corrected/commented:
line 34: abbreviation for “intensive care units (ICUs)”
Table 1: for colistin susceptibility testing using agar diffusion test see
“Uwizeyimana, J.D. et al., 2020. Ann.Lab.Med. 40: 306-311.
line 215: The (COS) clone as a variant of P.aeruginosa ST277 is wide-spread in Brazil. In the manuscript its “in-vitro phenotype” is emphasized as a specific characteristic. According to the comment in the previous review the question was raised how this term “in-vitro phenotype” is defined and differs from e.g. an “in-vivo phenotype”? Authors have not yet responded to the question, but are asked to provide a definition/explanation of this non-typical term.
Author Response
Answer to reviewer
line 34: abbreviation for “intensive care units (ICUs)”
Thanks for the suggestion. Text was changed.
Table 1: for colistin susceptibility testing using agar diffusion test see
“Uwizeyimana, J.D. et al., 2020. Ann.Lab.Med. 40: 306-311.
Thanks for the commentary, but we used broth microdilution (instead of agar dilution) for colistin susceptibility testing. Text was not changed.
line 215: The (COS) clone as a variant of P.aeruginosa ST277 is wide-spread in Brazil. In the manuscript its “in-vitro phenotype” is emphasized as a specific characteristic. According to the comment in the previous review the question was raised how this term “in-vitro phenotype” is defined and differs from e.g. an “in-vivo phenotype”? Authors have not yet responded to the question, but are asked to provide a definition/explanation of this non-typical term.
Thanks for the question. In fact, we mean that it was the phenotype observed in vitro. Since this term is not common, we decided to remove this word. Text was changed.